# A Single Dose of Beetroot Juice Does Not Change Blood Pressure Response Mediated by Acute Aerobic Exercise in Hypertensive Postmenopausal Women

**DOI:** 10.3390/nu11061327

**Published:** 2019-06-13

**Authors:** Ana Luiza Amaral, Igor M. Mariano, Victor Hugo V. Carrijo, Tállita Cristina F. de Souza, Jaqueline P. Batista, Anne M. Mendonça, Adriele V. de Souza, Douglas C. Caixeta, Renata R. Teixeira, Foued S. Espindola, Erick P. de Oliveira, Guilherme M. Puga

**Affiliations:** 1Laboratory of Cardiorespiratory and Metabolic Physiology, Federal University of Uberlândia, Uberlândia, MG 38400-678, Brazil; anaribeiro.am@gmail.com (A.L.A.); igormmariano@gmail.com (I.M.M.); vilarinhovictorh@gmail.com (V.H.V.C.); tallita_crystina@hotmail.com (T.C.F.d.S.); jaquebpontes@gmail.com (J.P.B.); annemarques.m@hotmail.com (A.M.M.); 2School of Medicine, Federal University of Uberlândia, Uberlândia, MG 38400-902, Brazil; erick_po@yahoo.com.br; 3Department of Food and Human Nutritional Sciences, University of Manitoba, Winnipeg, MB R3T 2N2, Canada; 4Canadian Centre for Agri-Food Research in Health and Medicine, St. Boniface Hospital Research Centre, Winnipeg, MB R2H 2A6, Canada; 5Laboratory of Biochemistry and Molecular Biology, Institute of Biotechnology, Federal University of Uberlândia, Uberlândia, MG 38400-902, Brazil; adriele_vds@hotmail.com (A.V.d.S.); caixetadoug@gmail.com (D.C.C.); rolandteixeira@yahoo.com (R.R.T.); fsespindola@gmail.com (F.S.E.)

**Keywords:** nitrite, nitric oxide, hypertension, menopause, Post Exercise Hypotension

## Abstract

Objective: To verify if acute intake of beetroot juice potentiates post-exercise hypotension (PEH) in hypertensive postmenopausal women. Methods: Thirteen hypertensive postmenopausal women (58.1 ± 4.62 years and 27.4 ± 4.25 kg/m²) were recruited to participate in three experimental sessions, taking three different beverages: Beetroot juice (BJ), placebo nitrate-depleted BJ (PLA), and orange flavored non-caloric drink (OFD). The participants performed moderate aerobic exercise training on a treadmill, at 65–70% of heart rate reserve (HRR), for 40 min. After an overnight fast, the protocol started at 07h when the first resting blood pressure (BP) was measured. The beverage was ingested at 07h30 and BP was monitored until the exercise training started, at 09h30. After the end of the exercise session, BP was measured every 15 min over a 90-min period. Saliva samples were collected at rest, immediately before and after exercise, and 90 min after exercise for nitrite (NO_2_^−^) analysis. Results: There was an increase in salivary NO_2_^−^ with BJ intake when compared to OFD and PLA. A slight increase in salivary NO_2_^−^ was observed with PLA when compared to OFD (*p* < 0.05), however, PLA resulted in lower salivary NO_2_^−^ when compared to BJ (*p* < 0.001). There were no changes in salivary NO_2_^−^ with the OFD. Systolic and diastolic BP decreased (*p* < 0.001) on all post exercise time points after all interventions, with no difference between the three beverages. Conclusion: Acute BJ intake does not change PEH responses in hypertensive postmenopausal women, even though there is an increase in salivary NO_2_^−^.

## 1. Introduction

There are several physiological changes in women’s body during climacteric period, including reduction of estrogen levels and menstruation cessation (menopause). As estrogen is a cardioprotective hormone [1], subsequent to its reduction, there is an increased risk of developing cardiometabolic diseases, due to reduced vasodilatation, increased blood pressure (BP), oxidative stress, and inflammation [2].

Hypertension is one of the most prevalent cardiometabolic diseases after menopause [3], and pharmacological therapy is the most common treatment approach. However, changes in lifestyle, such as developing healthy dietary habits and performing exercise regularly, are important factors for hypertension prevention and control [3]. The practice of regular physical exercise can reduce resting BP chronically, and induce post-exercise hypotension (PEH), which is a reduction in BP below resting values after exercise training. This phenomenon is clinically relevant in the prevention and treatment of cardiovascular diseases, and in the reduction of cardiovascular events [4]. This cardiovascular regulatory response has been related to endothelium-derived nitric oxide (NO) [4,5]. Exercise-induced shear stress promotes the release of calcium by endothelial cells, from which the calcium binds to calmodulin and stimulates endothelial nitric oxide synthase (eNOS) to reduce L-arginine to NO, where NO is directly involved in vascular tone regulation and homeostasis [5].

During exercise, NO release induces vasodilation, increasing blood flow to skeletal muscles and regulating blood pressure [5]. In addition, consumption of foods containing inorganic nitrate (NO_3_^−^) can increase NO bioavailability. This increase occurs by the initial conversion of NO_3_^−^ into nitrite (NO_2_^−^) in the mouth and stomach, through non-enzymatic reactions. The NO_2_^−^ is then released into the systemic circulation for subsequent NO production [6], leading to improvements in hemodynamic regulation. Therefore, consumption of foods rich in NO_3_^−^ may be a good strategy to improve BP regulation in target populations with an increased incidence of cardiometabolic diseases, especially when associated with exercise [7,8,9,10,11,12], since these two pathways of NO production can complement each other [6].

A common food that has a high NO_3_^−^ concentration is beetroot, and some effects observed with beetroot juice (BJ) consumption are related to improvement in blood flow, vasodilation, and reduction of BP [13,14,15]. The majority of studies that investigated BJ ingestion are related to sports and exercise performance [16,17,18,19], and only a few studies explored BJ consumption in parameters of health, in non-athletic populations, such as evaluating hemodynamic parameters in patients with chronic cardiorespiratory diseases [16,20]. These results are still inconclusive, since one did not find PEH with BJ consumption [20], and the other found PEH only in diastolic BP (DBP) [16]. Moreover, there is a lack of studies evaluating BJ benefits for hypertensive post-menopausal women. This population is of interest because there is a reduction in NO production, both due to hypertension and decreased estrogen levels.

Therefore, the purpose of this study was to verify if BJ ingestion has additional effects on PEH after one aerobic exercise session, and its relationship with salivary NO_2_^−^ in hypertensive postmenopausal women. Our hypothesis was that BJ would improve blood pressure response after exercise in hypertensive postmenopausal women, due to an increase in the NO pathway, measured by salivary NO_2_^−^ levels. 

## 2. Materials and Methods 

### 2.1. Participants

The study intervention was conducted between June and September 2018, at the Laboratory of Cardiorespiratory and Metabolic Physiology of the Federal University of Uberlândia, Uberlândia, MG, Brazil. This study design was approved by the local Ethics Committee (70104717.0.0000.5152) and registered at Clinicaltrials.gov (NCT03620227). All participants agreed and signed an informed consent form prior to admission.

For the inclusion criteria, participants were required to be: in post menopause (amenorrhea for at least 12 months and [FSH] > 40 mIU/mL); diagnosed with hypertension, according to the 7th Brazilian Arterial Hypertension Directive [3], which is defined by baseline blood pressure values greater than or equal to 140 mmHg for systolic BP (SBP) and 90 mmHg for DBP; aged between 50 and 70 years, and able to perform exercises on a treadmill. The exclusion criteria consisted of: use of hormonal therapy; history of food allergies that could compromise the study; sensitivity to NO_3_^−^; history of stroke or acute myocardial infarction; diagnosis of Diabetes Mellitus; and smoking habits. Although all the volunteers used antihypertensive drugs, they were excluded if they were taking drugs of the β-blocker class. All participants went through a cardiological evaluation with a specialist before the intervention, obtaining a certificate to attest individual suitability for exercise practice. Exclusion criteria also applied to volunteers who failed to perform the protocol test, with some of the reasons, including intolerance to the exercise program, inability to ingest the juice, or inability to go through the fasting time. 

The number of volunteers required for this trial was calculated considering BP variation caused by BJ relative to placebo (PLA) as the main variable, with a variation of 5 ± 4 mmHg defined as an acceptable effect [13]. Using online software (OpenEpi), considering a bilateral 95% confidence interval, and power analysis of 80%, a minimum of 11 people were needed for this study. After the calculation, 15 volunteers were recruited to participate in this study. Two out of the 15 volunteers were excluded from the study for inability to adapt to the study protocol, due to an abrupt decrease in blood pressure followed by initial syncope as of 15 min after the beginning of the first exercise session. One of the volunteers had consumed BJ and the other PLA.

The study volunteers answered both an anamnesis and a physical activity questionnaire (IPAQ short version). Anthropometric measures included: body mass (Filizola electronic scale); height (fixed stadiometer Sanny); abdominal circumference (Filizola inelastic tape); and body composition (bioimpedance Inbody 230, Seoul, Coreia do Sul), assessed as previously described [21]. For the body composition measurement, all participants were instructed not to perform vigorous physical exercise 24 h before the test and to avoid alcohol and caffeine consumption 72 h before the test.

### 2.2. Study Design

The present study was a double blind randomized, placebo-controlled, and crossover trial. The intervention comprised a total of three visits with a minimum wash-out interval of five days in between visits. In each visit, one of the following beverages was taken: Placebo (PLA), non-caloric orange flavor drink (OFD), or Beetroot Juice (BJ). Volunteers arrived in the laboratory at 07:00. and left at 11:40. Figure 1 illustrates the experimental design of the sessions.

During all intervention sessions, the volunteers arrived at 07:00, after eight hours of an overnight fast. BP was measured after 20 min of rest in a sitting position (07:20), every 30 min after the beverage intake until the beginning of the exercise (08:00, 08:30, and 09:00), and every 15 min for 90 min after the exercise session. Unstimulated saliva samples were collected after the 20 min of rest (07:20), immediately before (09:30) and after (10:10) exercise, and 90 min after the exercise session was finished (11:40). Heart rate (HR) was measured during the 20 min of rest (between 07:00 and 07:20), during the exercise, and during the 90 min after the exercise session (Polar^®^ RS800CX). The beverage intake took place 10 min after the rest period (07:30), allowing 15 min for the consumption of all the beverage content. The exercise session was 40 min long (from 09:30 to 10:10).

During the sessions, the volunteers were allowed to drink water, but no other drink or food ingestion was permitted. They were instructed to avoid foods and drinks rich in NO_3_^−^ 24 h before the sessions and received a list with the following consumption restrictions: Green vegetables (amaranth, lettuce, cabbage, spinach, broccoli, celery, cauliflower, Chinese radish), beetroot or its juice, sausage, salami, ham, turkey breast, coffee, energy drinks, soft drinks, alcoholic beverages, and to avoid the use of mouthwashes. Before starting each intervention session, volunteers were questioned about these items. 

### 2.3. Physical Exercise

The exercise consisted of 40 min of continuous moderate intensity aerobic exercise on a treadmill, allowing the first five minutes to warm-up and the last two minutes to cool down. The treadmill speed could reach 5.5 km/h and the intensity increase was imposed by inclining the treadmill until the volunteer reached the zone between 65% and 70% of HR reserve (HRR) [22]. For the HRR calculation, we used the formula: maximum HR—resting HR. For the participants resting HR, we considered the minimum HR measured during the initial 20 min of rest on the first intervention day, and the maximum HR was estimated by the formula: 220 − age.

HR was monitored during the exercise and the Borg Scale [23] was used to assess the subjective perception of exertion (RPE) for both dyspnea and lower limb fatigue. The measurements for HR and RPE were assessed every two minutes. Whenever the HR was found to be outside of the stipulated zone, the exercise load was readjusted.

### 2.4. Intake of Beetroot Juice and Placebo

The intervention included three different juices: BJ, PLA and OFD, with one beverage assigned to each intervention session. The order of beverage intake was randomly assigned for each of the volunteers using a randomized block design through the website (https://www.random.org/lists/). For the randomization, codes were assigned to each volunteer, each beverage, and each session. Thereafter, the beverage each volunteer would drink on the first session, second and third sessions were randomly allocated in this order. A researcher who did not participate in the data collection process was responsible for assigning a beverage to each code and blinding the drinks, adding each drink to its respective bottle labeled with the volunteer’s name. 

The BJ had 20.78 mmol/kg of NO_3_^−^ and was prepared using 35 mL of NO_3_^−^ concentrated beetroot juice containing 400 mg of NO_3_^−^ (Beet-It Sport Shot, James White Drinks Ltd., Ipswich, UK), which was diluted in 315 mL of distilled water with 6 g of non-caloric orange juice flavored powder (Clight, Mondelez International, Inc., São Paulo, Brazil), totalizing 350 mL of juice. The PLA had 3.86 mmol/kg of NO_3_^−^ and was prepared by filtering the BJ on an ion exchange resin, capable of depleting the NO_3_^−^ (PA101 OH-, Permution^®^) [24], similarly to a previous report [25]. Lastly, the OFD was a non-caloric orange flavored drink, prepared using three grams of orange juice powder diluted in 350 mL of distilled water. BJ and PLA were identical in taste, while OFD was slightly different. Each volunteer received the designated drink in a sealed bottle with a lid and a dark straw, making it impossible to visualize or smell the bottle content. The study participants had 15 min to drink the entire beverage.

### 2.5. Measurements of Blood Pressure

BP was measured in a sitting position, without visual or sound stimulation, using OMRON^®^ BP HEM-7113 automatic monitors. Prior to the BP measurement, 20 min of rest was required. BP was measured three times, always in the left arm, and the mean of these measures was calculated for data analysis.

### 2.6. Salivary Samples Collection and Analysis 

Saliva samples were collected using the spit method [26]. All samples were centrifuged at 3000 rpm for 15 min, the supernatant was separated and stored at −80 °C until analysis. The NO_2_^−^ concentration was used to estimate the bioavailability of NO by the Griess method [27].

### 2.7. Statistical Analysis 

Statistical analyses were performed while researchers were blinded regarding the ingested beverage. Results are presented as mean ± standard deviation. The Shapiro-Wilk test was applied to verify data normality, and two-way ANOVA was used to analyze differences between the time points (pre and post exercise) and the treatments (PLA, OFD, BJ), using the Bonferroni post hoc when necessary. The area under the curve (AUC) by the trapezoidal method was used to compare the temporal changes of BP and salivary NO_2_^−^, separately. One-way ANOVA was used for AUC and exercise analysis, and a Pearson correlation was used to assess BP and NO_2_^−^ effects. Statistical significance was set at *p* < 0.05. All analyses were performed using SPSS version 20 (IBM SPSS, Chicago, IL, USA) and GraphPad Prism 6 (GraphPad Prism Inc., San Diego, CA, USA).

## 3. Results

This study was completed by 13 postmenopausal women who were overweight, medically treated for hypertension, and physically active. The general characteristics of the volunteers are described in Table 1. There was no difference in exercise intensity between the sessions. For PLA, OFD, and BJ sessions, the HR mean was 125.6 ± 7.2; 126.1 ± 7.8; 126.7 ± 7.4 bpm (*p* = 0.658); the treadmill inclination was 3.4 ± 3.0; 3.5 ± 2.7; 3.3 ± 2.7% (*p* = 0.715); the RPE of dyspnea was 3.7 ± 0.85; 3.4 ± 1.0; 3.4 ± 1.0 (*p* = 0.328); and the RPE of lower limb fatigue was 4.1 ± 1.2; 3.9 ± 1.12; 3.8 ± 1.1 (*p* = 0.642), respectively.

Figure 2 illustrates the results for salivary NO_2_^−^. Participants had similar values during the sessions for salivary NO_2_^−^ at rest. Before exercise, salivary NO_2_^−^ was slightly increased (1.0 mM) by PLA and greatly increased by BJ (2.6 mM) (*p* < 0.05), while the values remained at a low level with OFD intake (0.1 mM). BJ lead to the highest salivary NO_2_^−^ level (3.3 mM) immediately after exercise when compared to the other beverages (PLA 0.9; OFD 0.1 mM) (*p* < 0.01). Furthermore, salivary NO_2_^−^ remained increased for up to 90 min post exercise with BJ intake when compared to rest (REST = 0.1 ± 0.1; POST 0′ = 3.3 ± 1.3; POST 90′ = 2.5 ± 1.1 mM; *p* < 0.001), but with PLA salivary NO_2_^−^ was only increased until immediately after exercise when compared to rest (REST = 0.2 ± 0.1; POST 0′ = 0.9 ± 0.6; POST 90′ = 0.7 ± 0.4 mM; *p* = 0.011). The AUC for NO_2_^−^ response over time was the highest with BJ when compared to both PLA and OFD (*p* < 0.01), however the response was still higher with PLA when compared to OFD (*p* = 0.037). There was no correlation between salivary NO_2_^−^ and BP (SBP *p* = 0.749, *r* = 0.053; DBP *p* = 0.618, *r* = −0.082). 

Figure 3 shows the SBP, DBP and HR responses throughout the experimental sessions. SBP increased after beverage consumption when compared to rest (*p* = 0.001) and decreased after exercise when compared to both rest and post ingestion time points. DBP also increased after beverage intake (*p* = 0.001) and decreased in the three following time points after exercise when compared to rest (*p* = 0.005). All DBP values decreased after exercise when compared to the pre-exercise value (*p* = 0.001). An increase in HR was observed after exercise when compared to rest (*p* = 0.001). There was no difference in SBP, DBP or HR between the three experimental sessions (*p* = 1.000).

Figure 4 demonstrates the BP variation after exercise in comparison with the BP measured at rest, 30 min before the exercise (at 9:00 am, as shown in Figure 1) and its respective AUC. SBP and DBP decreased in all evaluated time points post-exercise (*p* < 0.001). No difference for the AUC was observed among sessions (*p* = 1.000).

## 4. Discussion

The hypothesis underlying this study was that an increase in NO_2_^−^, due to BJ intake would enhance NO bioavailability, consequently increasing vasodilation and improving BP reduction mediated by exercise in hypertensive postmenopausal women. The main findings of the study were that a moderate-intensity aerobic exercise session was sufficient to cause PEH, and that acute BJ intake did not lead to additional effects on PEH, even though there was an increase in salivary NO_2_^−^, which could result in enhanced NO bioavailability.

To the best of our knowledge, this is the first study with hypertensive postmenopausal women analyzing acute intake of BJ and its influence on BP. Several other interventions assessing effects of BJ intake have been completed with different populations. Previous studies have investigated chronic [7,8,9,10,11,12,28] and acute BJ ingestion [16,20,28,29,30], in athletes [17,18,19], in healthy non-athletes [31,32], and in patients in disease states [7,8,9,10,11,12,16,20]. BJ consumption has been analyzed combined with exercise [28,33] and without exercise [34]. Usually, the primary objective of these studies is to evaluate the oxygen consumption and blood flow [28], associated with the analysis of plasma or salivary NO_2_^−^. Although BJ supplementation may cause a significant reduction in BP and this is usually evaluated as a secondary outcome, we can still cross-compare these studies with ours.

A recent review [6] discussed different study designs for BJ intervention and BP response in non-athlete populations. A meta-analysis showed that elderly people had less accentuated hypotensive responses [35], however, this is not a consensus, since another meta-analysis [13] showed an opposite effect. Other characteristics that may influence these results are both sex (women have less evident responses) and BP level at baseline. In this sense, the present study included middle age to elderly women (58.1 ± 4.6 years), which despite being hypertensive, had medically controlled BP at baseline. Therefore, these characteristics may have masked the hypotensive effects of BJ.

In a previous study with hypertensive patients with chronic obstructive pulmonary disease [16], there was a hypotensive effect in DBP three hours after BJ ingestion containing 12.9 mmol of NO_3_^−^. Different from the present study, they did not assess BJ consumption in combination with exercise, they used a lower dose of NO_3_^−^ in BJ, and they used a population with compromised cardiorespiratory capacity. In another study [20], patients with insufficient cardiac output using antihypertensive drugs consumed 140 mL of BJ containing 11.2 mmol of NO_3_^−^ and performed a 6 min walk, a maximal strength test, and a fatigue test. Despite evaluating BP post-exercise, they did not find a hypotensive effect associated with exercise or BJ supplementation. It should be noted that this evaluation occurred only 10 min after exercise, and the exercise protocol had more similarities to resistance exercise of high relative intensity. These characteristics differ greatly from the present study and may explain the absence of PEH, since this effect is better reported in post-aerobic exercise [36]. The amount of NO_3_^−^ used in our study was based on previous studies investigating different doses of beetroot/NO_3_^−^ and blood pressure responses [37]. Wylie and colleagues [37] found that consumption of NO_3_^−^ of up to 8.4 mmol had dose dependent lowering effects on BP, but higher doses had no additional effects on BP. Therefore, the dose used in our study should be enough to increase NO availability and increase blood flow.

After exercise, local vasodilator mechanisms contribute to BP reduction, leading to a sustained vasodilatation response [4]. There are studies suggesting that NO released by endothelium could lead to this effect, even though this is not the main cause of PEH [16]. Thus, nitric oxide could be considered a potential contributor to sustained post-exercise vasodilatation and could be involved in PEH. It is known that shear stress and activation of endothelial receptors cause NO release by eNOS and the capacity of this pathway depends on the interaction of neuropeptide and mechanical stimuli [38]. PEH is more evident after moderate intensity aerobic exercise in people with hypertension [39]. Considering the initial level of BP as one of the factors that determines the magnitude of PEH [39,40], the use of antihypertensive drugs before exercise could mitigate this response. However, we found a PEH response even with antihypertensive drug use, which highlights the importance of exercise even in hypertensive patients under pharmacological therapy. 

Additionally, the local vasodilatory response is not the only mechanism leading to PEH. Halliwill et al. [4] highlighted the possible explanations to PEH as: an adjustment of the baroreflex control to maintain a lower BP after exercise; a reduction in sympathetic nerve activity; a thermoregulatory readjustment; a vasodilation caused by release of NO and prostaglandins, which also cause reduction in α-adrenergic sensitivity; and a vasodilation caused by the action of histamines. It is possible that one or more of these pathways could be the main cause of PEH rather than NO bioavailability. Interestingly, BJ has a notable influence on the bioavailability of NO, and although this pathway does not seem to be the most important cause of PEH [4], some studies have shown that BJ intake can decrease BP [16,28,33,37]. Therefore, there is a possibility that increasing NO bioavailability by BJ consumption could change BP response. However, when considered along with other mechanisms causing PEH, there might be an attenuation or inhibition of other pathways involved that do not allow a significant additional effect in BP reduction.

In postmenopausal women, there is an increase in oxidative stress, due to the reduction in estrogen synthesis, impairing both vascular cell integrity and activity of antioxidant enzymes [41]. Furthermore, the impairment of oxidative balance has an important role in the pathogenesis of hypertension and may lead to impaired endothelium-dependent relaxation [42]. This decrease in relaxation capacity is associated with decreased NO production, increased cyclooxygenase-2 (COX-2) expression, and increased nicotinamide adenine dinucleotide phosphate oxidase (NADPH oxidase), of which COX-2 and NADPH oxidase are involved in stimulating the production of reactive oxygen species [42]. Thus, hypertensive women after menopause, have compromised blood vessel integrity, leading to endothelial dysfunction [42]. Ingestion of BJ [43] could be a clinically valid strategy to reduce the oxidative imbalance in this population, to increase the activity of antioxidant enzymes and blood flow, and to improve vascular relaxation dependent on shear stress derived from exercise. 

A study [25] compared the responses to BJ intake in plasma, saliva, and urine and found that saliva is more sensitive to NO_3_^−^, with the NO_2_^−^ concentration increasing seven times in saliva, three times in plasma, and four times in urine when compared to placebo. In agreement with these findings, in the current study there was a slight increase in salivary NO_2_^−^ with PLA intake, showing that even with the low concentration of NO_3_^−^ present in PLA, saliva is sensitive enough to respond to this stimulus [25]. This is likely due to the initial conversion of NO_3_^−^ to NO_2_^−^ that occurs in the mouth by salivary components [25,44], especially by oral microbiome [45]. The bacteria genus *Veillonella* spp. is of importance, increasing the conversion of NO_3_^−^ to NO_2_^−^ in the oral cavity, thus, assisting with the reintroduction of NO_2_^−^/NO_3_^−^ in the pathway of NO production [45]. Recently, the clinical importance of the oral microbiome has been demonstrated in a hypertensive population by the use of antibacterial mouthwashes, which decreased the oral reduction of NO_3_^−^ and increased blood pressure in this population [46] 

Based on our results, there are other possibilities to explain the lack of an additional effect on PEH with BJ ingestion. The literature shows that the amount of NO_3_^−^ consumed is sufficient to cause a hypotensive response in healthy individuals [37], however, the NO_3_^−^ dose response could vary among different populations. Therefore, this dose may not have been enough for hypertensive postmenopausal women. Although there are indications that increased bioavailability of NO_2_^−^ is associated with the increased bioavailability of NO [47,48], this conversion may not have been satisfactory to generate enough NO to potentiate vasodilation. In addition, the endothelium status of the participants is not known. As all participants have been diagnosed with hypertension for more than one year, there may be a deficiency in the production of eNOS, which could reduce NO production [49]. Thus, it is important to evaluate the NO response, to determine NO_3_^−^ dose responses in different populations, and evaluate chronic consumption of BJ. Although there are no studies evaluating the interaction of antihypertensives and blood pressure response with exercise [50], it is important to consider that the study volunteers had pharmacologically controlled hypertension, and these drugs could influence blood pressure response after exercise. Furthermore, the bioavailability of NO may not be the main cause of PEH, and other mechanisms causing PEH were not evaluated in the present study, of which we highlight the activity of baroreflex and vasodilation caused by other substances, such as histamine and prostaglandins [4].

This study had limitations, since blood flow was not measured, and BJ intake and BP measurements were assessed in a short term, thus, our results cannot be extrapolated to chronic settings. The study participants were taking different anti-hypertensive medication, which could mask the effect of NO_3_^−^ on blood pressure. Our results suggest that moderate aerobic exercise is a good strategy to induce PEH and may be helpful for hypertension treatment. However, only one dose of BJ is not sufficient to cause an additional effect on PEH.

## 5. Conclusions

In conclusion, acute intake of beetroot juice does not change BP response mediated by moderate intensity aerobic exercise in hypertensive postmenopausal women, even though there is an increase in the bioavailability of salivary NO_2_^−^/NO.

## Figures and Tables

**Figure 1 nutrients-11-01327-f001:**
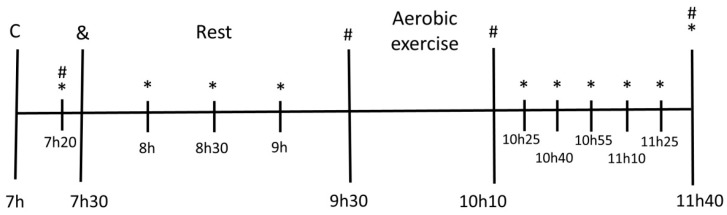
Experimental design of the sessions. (C) Arrival at the laboratory; (*) Blood pressure measurement; (#) Saliva sample collection; (&) Beverage intake.

**Figure 2 nutrients-11-01327-f002:**
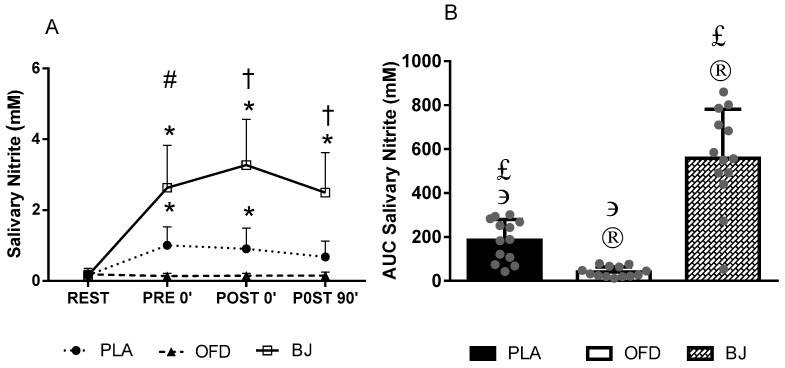
Salivary NO_2_^−^ values from point to point (**A**) and values for the area under the curve (AUC) (**B**). Placebo NO_3_^−^ depleted beetroot juice (PLA); Non-caloric orange flavored drink (OFD); Beetroot juice (BJ). (#) significantly different between all sessions; (†) significantly different when comparing BJ with both PLA and OFD; (*) significantly different when compared to rest (REST); (®) significantly different when compared to PLA; (϶) significantly different when compared to BJ; (£) significantly different when compared to OFD.

**Figure 3 nutrients-11-01327-f003:**
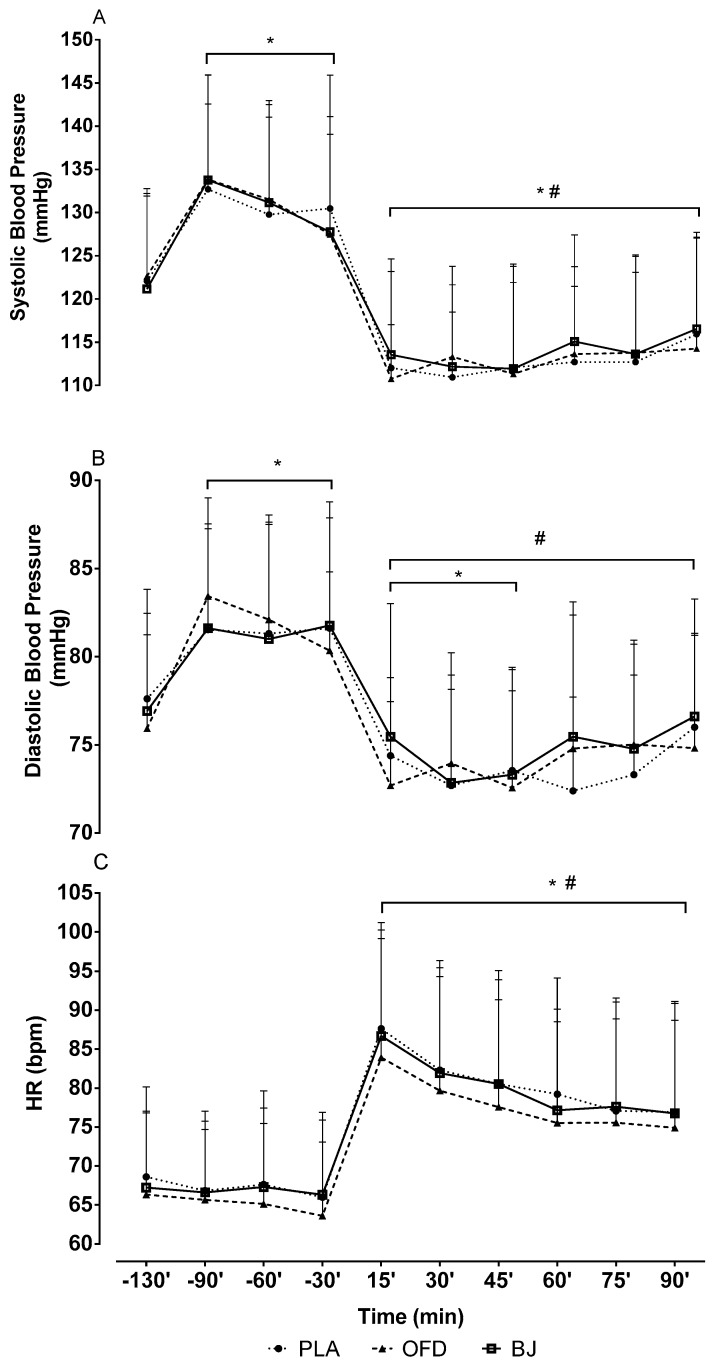
Systolic blood pressure (**A**); diastolic blood pressure (**B**) and point-to-point heart rate (**C**). Placebo NO_3_^−^ depleted beetroot juice (PLA); Non-caloric orange flavor drink (OFD); Beet juice (BJ). The negative time values refer to measurements before exercise. The positive time values refer to measurements after the exercise. (*) significantly different when compared to the −130′ timepoint (resting pre-ingestion); (#) significantly different when compared to the −90’, −60’, −30’ timepoints (post juice intake).

**Figure 4 nutrients-11-01327-f004:**
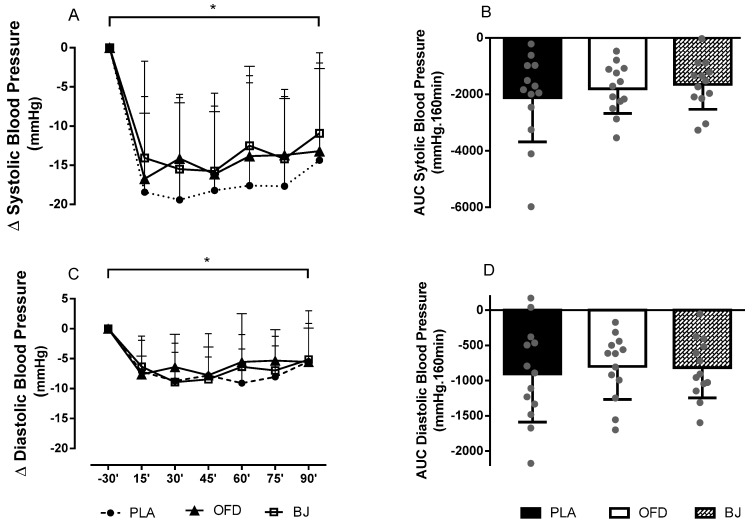
Δ values for systolic blood pressure variation (**A**) and diastolic blood pressure variation (**C**), area under the curve (AUC) for systolic blood pressure (**B**) and for diastolic blood pressure (**D**). Placebo NO_3_^−^ depleted beetroot juice (PLA); Non-caloric orange flavor drink (OFD); Beetroot juice (BJ). The negative time values refer to measurements before exercise. The positive time values refer to measurements after the exercise. (*) significantly different when compared to the −30’ point (pre-exercise).

**Table 1 nutrients-11-01327-t001:** General characteristics of the participants. BMI: Body mass index. The general characteristics values are shown as mean ± standard deviation (SD), physical activity level and drugs are shown as: number of subjects (n) and percentage of the total number of subjects (%).

**General Characteristics**	**(Mean ± SD)**
Age (years)	58.1 ± 4.6
Body mass (kg)	69.9 ± 9.2
Height (m)	1.57 ± 0.05
BMI (kg/m²)	27.4 ± 4.2
Waist circumference (cm)	92.9 ± 11.7
Body fat (%)	37.3 ± 6.2
Fat mass (kg)	26.1 ± 6.9
Lean mass (kg)	29.9 ± 9.2
**Physical activity level**	**(n (%))**
Very Active	2 (15%)
Active	8 (62%)
Irregularly Active	3 (23%)
**Drugs**	**(n (%))**
Angiotensin 1 Receptor Blockers + Diuretic	6 (46%)
Angiotensin 1 Receptor Blockers	4 (31%)
Diuretic	1 (8%)
Angiotensin Converting Enzyme Inhibitor	1 (8%)
Angiotensin Converting Enzyme Inhibitor + Diuretic	1 (8%)
Statins	3 (23%)
Levothyroxine	4 (31%)

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
