# Peer review of "A Single Dose of Beetroot Juice Does Not Change Blood Pressure Response Mediated by Acute Aerobic Exercise in Hypertensive Postmenopausal Women"

_nutrients, 2019, doi:10.3390/nu11061327_

Reviewer 1 Report

Summary

Amaral et al of this manuscript investigated the hypothesis that that beetroot juice (BJ) would improve post-exercise hypotension (PHE) due to an increase in NO pathway, measured by salivary NO2 levels in post hypertensive 26 postmenopausal women. The authors found that increased salivary NO2 with BJ and no change in PHE responses.

Specific comments:

1.      Authors could have shown the maximum oxygen uptake (VO2 max) to determine the exercise capacity.

2.      Endothelial dysfunction is the major concern in hypertension, even increased NO2, dysfunctional endothelium may not compensating for NOS production to reduce SBP in PHE. Authors could have shown the flow-mediated dilation method to demonstrate endothelial dysfunction.

3.      Authors should include in the discussion about the dose of the BJ, if increase the doses may have an impact on SBP etc.

Author Response

Response to reviewers

Ref.:  ID nutrients-506762

A single dose of beetroot juice does not change blood pressure response mediated by acute aerobic exercise in hypertensive postmenopausal women: a cross-over, randomized, double-blind, placebo-controlled study

Dear Ms. Elisa Yuan

Assistant Editor, Nutrients 

We would like to thank you for the opportunity to revise and resubmit our manuscript.  The reviewers’ comments were very helpful to improve our paper. Please find below our answers to each comment. We performed a revision of the entire manuscript according to the reviewers’ suggestions which will hopefully improve the clarity and readability of the manuscript.

In order to make the changes more visible, all edits to the manuscript were highlighted with Track Changes in Word.

Response to Reviewer #1 Comments:

·     Point 1: “Authors could have shown the maximum oxygen uptake (VO2max.) to determine the exercise capacity”

Response: Thank you for the comment. We also believe that the evaluation of VO2max can be very important to determine the exercise capacity and to evaluate aerobic fitness. However, it was not our goal to evaluate neither the aerobic capacity nor exercise capacity. Instead, our goal was to investigate the post exercise blood pressure response. Therefore, the volunteers performed moderate intensity exercise for 40 minutes, with the intensity controlled by the heart rate response, which has a similar and linear relationship with the VO2. Either the relationship between heart rate and VO2with exercise intensity, or the use of only heat rate to control the aerobic moderate exercise performance are well established in the literature (Mass et al., 1989). We chose this exercise intensity (65-70% of heart rate reserve) because it is an exercise intensity that is also well established to promote the post exercise hypotension in hypertensive patients.

Maas, S.; Kok, M.L.; Westra, H.G.; Kemper, H.C.G. The validity of the use of heart rate in estimating oxygen consumption in static and in combined static/dynamic exercise. Ergonomics198932, 141–148.

·     Point 2: “Endothelial dysfunction is the major concern in hypertension, even increased NO2, dysfunctional endothelium may not compensating for NOS production to reduce SBP in PHE. Authors could have shown the flow-mediated dilation method to demonstrate endothelial dysfunction”

Response: Thank you for this comment. We agree that the major concern in hypertension is endothelium dysfunction, and in this case, the nitrate/nitrite supplementation cannot compensate for the impaired Nitric oxide production. However, several studies have shown that the intake of nitrate rich foods, such as beetroot juice, can increase the flow-mediated dilatation and local blood flow (Asgary et al., 2016; Lee et al., 2015) by using the same amount of nitrate and juice as we did. Therefore, we believe that this nitrate ingestion could improve the blood flow mediated by NO/nitrate/nitrite pathways. Unfortunately, we were not able to measure the flow-mediated dilation, and this is included as a limitation of our study at the end of the manuscript discussion.

·     Asgary, S.; Afshani, M.R.; Sahebkar, A.; Keshvari, M.; Taheri, M.; Jahanian, E.; Rafieian-Kopaei, M.; Malekian, F.; Sarrafzadegan, N. Improvement of hypertension, endothelial function and systemic inflammation following short-term supplementation with red beet (Beta vulgaris L.) juice: A randomized crossover pilot study. J. Hum. Hypertens.201630, 627–632.

·     Lee, J.-S.; Stebbins, C.L.; Jung, E.; Nho, H.; Kim, J.-K.; Chang, M.-J.; Choi, H.-M. Effects of chronic dietary nitrate supplementation on the hemodynamic response to dynamic exercise. Am. J. Physiol. Integr. Comp. Physiol.2015309, 459–466.

·     Point 3: “Authors should include in the discussion about the dose of the BJ, if increase the doses may have an impact on SBP etc.”

Response: Thank you for the comment. The study by Wylie et al. (2013) investigated the intake of different Nitrate (NO3-) doses and its relationship with blood pressure response. They found that doses higher than 8.4mmol and up to 16.8 mmol had no additional effect on blood pressure reduction. Therefore, we believe that the dose of nitrate from Beetroot juice that we used in this study was enough. We have included this information in the discussion session at the end of the 4thparagraph. 

·     Wylie, L.J.; Kelly, J.; Bailey, S.J.; Blackwell, J.R.; Skiba, P.F.; Winyard, P.G.; Jeukendrup, A.E.; Vanhatalo, A.; Jones, A.M. Beetroot juice and exercise: pharmacodynamic and dose-response relationships. J. Appl. Physiol.2013115, 325–336.

Reviewer 2 Report

Amaral et al. investigated whether a single dose of beetroot juice (dietary nitrate) would alter the post-exercise blood pressure response in hypertensive post-menopausal women.  Blood pressure typically decreases after a bout of exercise and this study examined whether dietary nitrate would increase this response.  On separate occasions, participants consumed three different beverages: nitrate-rich beetroot juice diluted with orange drink, nitrate-depleted beetroot juice diluted with orange drink, and a calorie-free orange drink.  The investigators compared salivary nitrite and blood pressure throughout a protocol that included waiting for entero-salivary circulation to convert nitrate to nitrite, a 40 min bout of aerobic treadmill exercise, and a recovery period.  Salivary nitrite increased following beetroot juice, but blood pressure responses were similar under all three study conditions.

The authors are to be complimented for undertaking research with this population, post-menopausal women are under-represented in health literature.  There are, however, a number of issues that detract from the work presented here.

Comments:

1)      Post-exercise hypotension is relevant in the prevention and treatment of cardiovascular disease.  However, large decreases in blood pressure following exercise can cause nausea and syncope, as noted in two of 15 participants in this study.  Similarly, high intensity interval studies report PEH as a consequence and sometimes a deterrent to exercise.  With that in mind, why would you be seeking to increase the hypotensive response to moderate intensity exercise?  

2)      The authors are referred to Kim-Shapiro and Gladwin, 2014 (1).  Nitrite conversion to NO is facilitated under conditions of ischemia and low pH, for example, under low oxygen tension, haemoglobin-linked pathways are stimulated (in a protective fashion).  Following exercise in hypertensive post-menopausal women, blood pressure is already low and participants are at rest, one would not expect an increase in nitrite conversion to NO under these conditions.

3)      This study included participants who take medication. Anti-hypertensive medications may act via vasodilatory pathways with parallel purpose to the NO3- - NO2- - NO pathway.  For example, inhibiting the renin-angiotensin system with angiotensin converting enzyme (ACE) inhibitors or angiotensin receptor blockers (ARBs) would have a vasodilating effect and this is likely to mask any effect of a dietary nitrate supplement.

Specific revisions:

I have a great deal of respect for the effort involved in writing a scientific paper in a language that is not one’s native language.  Please consider my comments constructive, rather than critical.  Throughout this paper, there are opportunities to make the message clearer by changing the wording and by removing distracting grammatical errors.  Sometimes I have suggested alternatives, but in many cases, I will just draw your attention to it.

4)      Lines 1-5: The title is long and convoluted.  Consider removing the study design from the title of the paper

5)      Line 24: wording “an acute”

6)      Line 26: wording “of BMI”

7)      Lines 27-28: wording: use “nitrate-depleted BJ” rather than BJ nitrate-depleted

8)      Line 36: wording: “was still lower”

9)      Line 44: wording: “climacteric”

10)   Line 46: wording: use “subsequent” rather than subsequently

11)   Line 49: wording: disease needs to be plural

12)   Line 51: wording: “adequate”

13)   Line 51: wording: “positively influence hypertension prevention” this is awkward phrasing

14)   Lines 52-54: wording: this could be phrased more clearly

15)   Line 54: please do not start a sentence with an abbreviation

16)   Line 57: the endothelial cells are responsive to shear stress (and other stimuli) and they release Calcium, which binds to Calmodulin, which in turn, stimulates eNOS.  Shear stress does not directly stimulate eNOS.

17)   Lines 58-59: wording “vascular structure control”

18)   Line 60: “blood flow control”?

19)   Line 62: wording: “distinct initial mechanism”

20)   Lines 62-65: this explanation could be tidier

21)   Line 65: ingestion of dietary nitrate may be a strategy…

22)   Lines 60 – 75: sometimes NO3- is used, sometimes nitrate is used.  Please be consistent (I'm aware that I have used both in this review document, however it is important to be consistent in the manuscript).

23)   Lines 69-75: While the purpose of this study is conveyed with reference to other studies involving beetroot juice, the literature has not been thoroughly reviewed.  Relevant work examining older adults and patients with cardiovascular disease have been overlooked, eg refs  (2-7) and at least 2 meta-analyses (8, 9)

24)   Lines 76-79: a point of difference in this study is that measures were taken in post-menopausal women.  This is a strength and should be emphasized in the purpose statement.

25)   Line 78: improve PEH is confusing.  To make it better means a further decrease in pressure?  Please ensure your hypothesis is clearly stated.

26)   Line 95: “could not use” instead please explain that they were excluded or this was an exclusion criteria  

27)   Line 116: “crossover, randomized, double blind, placebo-controlled” consider changing the order of these descriptors for clarity.

28)   Line 117: 5 days washout with 3 conditions, 10 days doesn’t add up?

29)   Line135: please consider a longer abstinence from nitrate rich foods in future studies

30)   Line 181: wording should be: statistical analyses were performed while researchers were blinded regarding the ingested beverage

31)   Table 1: some of your variables are n and % distribution, please add the heading where relevant

32)   Results: when a symbol separates two numbers, there should be a spaces on either side, eg 3.4 ± 3.0

33)   Line 197: wording “all sessions had similar values…” the sessions did not have values, rather the participants had similar values during the sessions

34)   Line 230-231: which resting measure was used?  There were four reported prior to the exercise.

35)   Line 245: “even though NO bioavailability was enhanced”.  This is debatable.  Salivary nitrite was elevated, but that does not necessarily translate into elevated NO bioavailability, which you have not measured.

36)   Line 247: This is the main strength of your study.  Great!

37)   Line 248-255: as with the introduction, there are important citations missing in this paragraph.  This is not a thorough summary of where the current study fits within the literature.

38)   Line 256-257: please position references consistently, either after “review” and “meta-analysis” or at the end of the sentences, but avoid switching between these tactics

39)   Line 258-259: opposing findings in the literature could be presented more clearly

40)   Line 262-263: wording “collaborated to mitigate”

41)   Line 265: 12.9 (space) mmoles of NO (subscript) 3 –

42)   Line 276: nitric oxide might contribute toward this effect, please be careful with overstatement

43)   Line 277: please avoid starting a sentence with an abbreviation

44)   Lines 284-285: Yes!  Well stated!

45)   Line 286: wording: “justifying”

46)   Line 287: wording

47)   Lines 287-293: Listing the possible mechanisms suggested by Halliwell suggests you are going to interpret these ideas and position your findings in context, you have not.

48)   Line 296: wording: “blockage or limitation”, please explain what you mean

49)   Line 299-300: please read Forstermann, 2010 (10) and Selemidis, 2008 (11).  The relationship between oxidative stress and antioxidant capacity is misrepresented by this statement.

50)   Line 302: “capacity” is incorrect here

51)   Lines 305-309: this is an important statement for your paper.  Please ensure concise wording (avoid supposedly) and make your message very clear (what do you mean by difficult responses?)

52)   Lines 310-316: please read Burleigh et al, 2018 (12) and Blekkenhorst, 2018 (13), for details on the contribution of the oral microbiome

53)   Lines 317-322: this is the crux of your discussion and should be discussed fully and referenced comprehensively

54)   Lines 327-329: these are author instructions that should have been deleted before submission

 Thank you for this opportunity to review your manuscript.  Best wishes.   

1.            Kim-Shapiro DB, Gladwin MT. Mechanisms of nitrite bioactivation. Nitric Oxide-Biology and Chemistry. 2014;38:58-68.

2.            Eggebeen J, Kim-Shapiro DB, Haykowsky M, Morgan TM, Basu S, Brubaker P, et al. One Week of Daily Dosing With Beetroot Juice Improves Submaximal Endurance and Blood Pressure in Older Patients With Heart Failure and Preserved Ejection Fraction. JACC-Heart Fail. 2016;4(6):428-37.

3.            Kenjale AA, Ham KL, Stabler T, Robbins JL, Johnson JL, VanBruggen M, et al. Dietary nitrate supplementation enhances exercise performance in peripheral arterial disease. J Appl Physiol. 2011;110(6):1582-91.

4.            Kelly J, Fulford J, Vanhatalo A, Blackwell JR, French O, Bailey SJ, et al. Effects of short-term dietary nitrate supplementation on blood pressure, O-2 uptake kinetics, and muscle and cognitive function in older adults. American Journal of Physiology-Regulatory Integrative and Comparative Physiology. 2013;304(2):R73-R83.

5.            Bock JM, Treichler DP, Norton SL, Ueda K, Hughes WE, Casey DP. Inorganic nitrate supplementation enhances functional capacity and lower-limb microvascular reactivity in patients with peripheral artery disease. Nitric Oxide. 2018.

6.            Siervo M, Oggioni C, Jakovljevic DG, Trenell M, Mathers JC, Houghton D, et al. Dietary nitrate does not affect physical activity or outcomes in healthy older adults in a randomized, cross-over trial. Nutr Res. 2016;36(12):1361-9.

7.            Woessner MN, VanBruggen MD, Pieper CF, Sloane R, Kraus WE, Gow AJ, et al. Beet the Best? Dietary Inorganic Nitrate to Augment Exercise Training in Lower Extremity Peripheral Artery Disease with Intermittent Claudication. Circ Res. 2018.

8.            Jackson JK, Patterson AJ, MacDonald-Wicks LK, Oldmeadow C, McEvoy MA. The role of inorganic nitrate and nitrite in cardiovascular disease risk factors: a systematic review and meta-analysis of human evidence. Nutr Rev. 2018;76(5):348-71.

9.            Stanaway L, Rutherfurd-Markwick K, Page R, Ali A. Performance and Health Benefits of Dietary Nitrate Supplementation in Older Adults: A Systematic Review. Nutrients. 2017;9(11):16.

10.          Forstermann U. Nitric oxide and oxidative stress in vascular disease. Pflugers Archiv-European Journal of Physiology. 2010;459(6):923-39.

11.          Selemidis S. Suppressing NADPH oxidase-dependent oxidative stress in the vasculature with nitric oxide donors. Clinical and Experimental Pharmacology and Physiology. 2008;35(11):1395-401.

12.          Burleigh MC, Liddle L, Monaghan C, Muggeridge DJ, Sculthorpe N, Butcher JP, et al. Salivary nitrite production is elevated in individuals with a higher abundance of oral nitrate-reducing bacteria. Free Radical Biology and Medicine. 2018;120:80-8.

13.          Blekkenhorst LC, Bondonno NP, Liu AH, Ward NC, Prince RL, Lewis JR, et al. Nitrate, the oral microbiome, and cardiovascular health: A systematic literature review of human and animal studies. Am J Clin Nutr. 2018;107(4):504-22.

Author Response

Response to reviewers

Ref.:  ID nutrients-506762

A single dose of beetroot juice does not change blood pressure response mediated by acute aerobic exercise in hypertensive postmenopausal women: a cross-over, randomized, double-blind, placebo-controlled study

Dear Ms. Elisa Yuan

Assistant Editor, Nutrients 

We would like to thank you for the opportunity to revise and resubmit our manuscript.  The reviewers’ comments were very helpful to improve our paper. Please find below our answers to each comment. We performed a revision of the entire manuscript according to the reviewers’ suggestions which will hopefully improve the clarity and readability of the manuscript.

In order to make the changes more visible, all edits to the manuscript were highlighted with Track Changes in Word.

Response to Reviewer #2 Comments:

·     Overall Point: “English translation can be improved.”

Response: Thank you for reviewing our manuscript and for all your thoughtful comments. A native English speaker has reviewed the English of the entire manuscript.

·     Point 1: “Post-exercise hypotension is relevant in the prevention and treatment of cardiovascular disease.  However, large decreases in blood pressure following exercise can cause nausea and syncope, as noted in two of 15 participants in this study.  Similarly, high intensity interval studies report PEH as a consequence and sometimes a deterrent to exercise.  With that in mind, why would you be seeking to increase the hypotensive response to moderate intensity exercise?”

Response: Thank you for the comment. This is an important point. Post exercise hypotension has been extensively studied and it is considered an important mechanism for the prevention and treatment of cardiovascular diseases. We agree that in some specific cases PEH might not have a positive effect, but in the majority of cases, lowering blood pressure response is considered a positive effect, particularly for hypertensive patients. Our goal was to investigate if beetroot juice ingestion would have an effect on post-aerobic exercise blood pressure response in hypertensive post menopause women, and moderate exercise is highly recommended for this population. Some studies have shown that moderate exercise (Halliwil et al., 2013; Brito et al., 2018) and beetroot ingestion (Lara et al., 2016) could increase blood flow and decrease blood pressure even in healthy young volunteers, but none of them reported this response as a negative effect. Furthermore, we believe that a large decrease in blood pressure could have a negative effect, however, to the best of our knowledge, such a negative effect has not been investigated or shown due to the association of moderate aerobic exercise and beetroot juice intake. Accordingly, we wanted to first identify if there were any additional effects on PEH by beetroot juice consumption, and then determine if post menopause hypertensive patients could benefit from this response to improve hypertension. Furthermore, it seems like the initial value of blood pressure determines the magnitude of post exercise hypotension, therefore the greater the initial blood pressure, the better the results on lowering the blood pressure after exercise (Brito et al., 2018). There are probably mechanisms, such as the baroreceptor reflex response, that do not allow the blood pressure to decrease below a safe point. In our study, two women could not complete the intervention because they got dizzy, probably due to an abrupt decrease in blood pressure. However, it is not clear whether this has happened due to: the beetroot juice or placebo ingestion, the association with a fasting state, the medication taken by the participants, the exercise intensity, or even the local temperature or the body temperature. In my experience at our laboratory, we have tested hundreds of normotensive and hypertensive volunteers and had never had this problem before. This cause and effect should be better explored. Moreover, our results show that the ingestion of beetroot juice does not change blood pressure response in pharmacologically treated hypertensive post-menopausal women.

·     Halliwill, J.R.; Buck, T.M.; Lacewell, A.N.; Romero, S.A. Postexercise hypotension and sustained postexercise vasodilatation: what happens after we exercise? Exp. Physiol.2013, 98, 7–18.

·     Brito, L.C.; Fecchio, R.Y.; Peçanha, T.; Andrade-Lima, A.; Halliwill, J.R. Forjaz, C.L.M. Postexercise hypotension as a clinical tool: a “single brick” in the wall. Journal of the American Society of  Hypertension. 2018, 12, 59-64.

·     Lara , J.; Ashor, A. W.; Oggioni, C.; Ahluwalia, A.; Mathers, J. C.; Siervo, M. Effects of inorganic nitrate and beetroot supplementation on endothelial function: a systematic review and meta-analysis. European Journal of Nutrition. 2016, 55, 451–459

·     Point 2: “The authors are referred to Kim-Shapiro and Gladwin, 2014.  Nitrite conversion to NO is facilitated under conditions of ischemia and low pH, for example, under low oxygen tension, haemoglobin-linked pathways are stimulated (in a protective fashion). Following exercise in hypertensive post-menopausal women, blood pressure is already low and participants are at rest, one would not expect an increase in nitrite conversion to NO under these conditions.”

Response: Thank you for the comment. This is another important point. The study from Kim-Shapiro e Gladwin (2014) showed that under conditions of ischemia and low pH, nitrate conversion to NO is facilitated. This is a physiological condition that occurs during exercise and can extend to after exercise as well. Further, after exercise women were at rest, and there are several physiological changes that will be different until the body reaches homeostasis during this phase. The shear stress and consequent NO release by eNOS occurs heavily during exercise, however, NO production can last longer after the end of exercise. This happens due to alterations such as increased blood flow and reduced pH when compared with the rest state before exercise (Froio et al., 2015). These conditions can also facilitate nitrite conversion to NO. Beetroot juice has a notable influence on the bioavailability of NO, and although this pathway does not seem to be the most important cause of post-exercise hypotension, some studies have shown that beetroot juice intake can decrease blood pressure. Therefore, we believe that there is a possibility that increasing NO bioavailability by beetroot juice consumption could change blood pressure response. However, when two or more conditions that cause post-exercise hypotension occur simultaneously, there might be an attenuation or inhibition of certain pathways that does not allow a significant additional effect in blood pressure reduction.

·     Kim-Shapiro, D.B.; Gladwin, M.T. Mechanisms of Nitrite Bioactivation. Nitric Oxide2014, 58–68.

·     Froio, G.; Dias, D.A.; Silva, E.; Painelli, V.D.S.; Sale, C.; Artioli, G.G.; Gualano, B.; Saunders, B. ( In ) Consistencies in Responses to Sodium Bicarbonate Supplementation: A Randomised , Repeated Measures , Counterbalanced and Double-Blind Study. 2015,284164409.

·     Point 3: “This study included participants who take medication. Anti-hypertensive medications may act via vasodilatory pathways with parallel purpose to the NO3- - NO2- - NO pathway.  For example, inhibiting the renin-angiotensin system with angiotensin converting enzyme (ACE) inhibitors or angiotensin receptor blockers (ARBs) would have a vasodilating effect and this is likely to mask any effect of a dietary nitrate supplement.”

Response:Thank you for the comment. We believe that this is an important point as well. The anti-hypertensive medication may influence blood pressure response after exercise, but there is no study that showed if this really occurs in all cases or how this can influence blood pressure response after exercise. There are several mechanisms that can decrease blood pressure after exercise, as discussed in the manuscript, but the literature showed post exercise hypotension even in patients under different anti-hypertensive drugs (Eggebeen et al, 2016; Ocampo et al, 2018). To minimize this effect, we tested each volunteer in the same condition during all three experimental sessions, taking into consideration the time and dose of drug consumed. Also, the volunteers were using the same drug for at least one-year straight. Moreover, when we analyzed the individual responses, we could not see any difference among volunteers who take different medications. Even with all these considerations, this could be one limitation of our study, accordingly, we have included this information with the limitations in the discussion session.

Eggebeen, J.; Kim-Shapiro, D.B.; Haykowsky, M.; Morgan, T.M.; Basu, S.; Brubaker, P.; Rejeski, J.; Kitzman, D.W. One Week of Daily Dosing With Beetroot Juice Improves Submaximal Endurance and Blood Pressure in Older Patients With Heart Failure and Preserved Ejection Fraction. JACC Hear. Fail.20164, 428–437.

Ocampo, D.A.B.; Paipilla, A.F.; Marín, E.; Vargas-Molina, S.; Petro, J.L.; Pérez-Idárraga, A. Dietary Nitrate from Beetroot Juice for Hypertension: A Systematic Review. Biomolecules2018, 1–12.

·     Point 4: “Lines 1-5: The title is long and convoluted.  Consider removing the study design from the title of the paper

Response: Thank you for the comment. We have changed the title as suggested.

·     Point 5: “Line 24: wording “an acute””

Response: We have adjusted the phrase for better clarity and readability.

·     Point 6: “Line 26: wording “of BMI””

Response: We have adjusted the phrase for better clarity and readability.

·     Point 7: “Lines 27-28: wording: use “nitrate-depleted BJ” rather than BJ nitrate-depleted”

Response: We have adjusted the phrase for better clarity and readability.

·     Point 8:  “Line 36: wording: “was still lower””

Response: We have adjusted the phrase for better clarity and readability.

 ·     Point 9: “Line 44: wording: “climacteric””

Response: We have adjusted the phrase for better clarity and readability.

 ·     Point 10: “Line 46: wording: use “subsequent” rather than subsequently”

Response: We have adjusted the phrase for better clarity and readability.

 ·     Point 11: “Line 49: wording: disease needs to be plural”

Response: We have adjusted the phrase for better clarity and readability.

 ·     Point 12: “Line 51: wording: “adequate””

Response: We have adjusted the phrase for better clarity and readability.

 ·     Point 13: “Line 51: wording: “positively influence hypertension prevention” this is awkward phrasing”

Response: We have adjusted the phrase for better clarity and readability.

 ·     Point 14: “Lines 52-54: wording: this could be phrased more clearly”

Response: We have adjusted the phrase for better clarity and readability.

 ·     Point 15: “Line 54: please do not start a sentence with an abbreviation”

Response: We have adjusted the phrase for better clarity and readability.

 ·     Point 16: “Line 57: the endothelial cells are responsive to shear stress (and other stimuli) and they release Calcium, which binds to Calmodulin, which in turn, stimulates eNOS.  Shear stress does not directly stimulate eNOS.”

Response: We have adjusted the phrase for better clarity and readability.

 ·     Point 17: “Lines 58-59: wording “vascular structure control””

Response: We have adjusted the phrase for better clarity and readability.

·     Point 18: “Line 60: blood flow control?”

Response: We have adjusted the phrase for better clarity and readability.

 ·     Point 19: “Line 62: wording: “distinct initial mechanism” ”

Response: We have adjusted the phrase for better clarity and readability.

 ·     Point 20: “Lines 62-65: this explanation could be tidier”

Response: We have adjusted the phrase for better clarity and readability.

 ·     Point 21: “Line 65: ingestion of dietary nitrate may be a strategy…”

Response: We have adjusted the phrase for better clarity and readability.

 ·     Point 22: “Lines 60 – 75: sometimes NO3-is used, sometimes nitrate is used. Please be consistent (I'm aware that I have used both in this review document, however it is important to be consistent in the manuscript).”

Response: Thank you for the comment. We have made the suggested adjustments, and the entire text is now consistent. 

·     Point 23: “Lines 69-75: While the purpose of this study is conveyed with reference to other studies involving beetroot juice, the literature has not been thoroughly reviewed.  Relevant work examining older adults and patients with cardiovascular disease have been overlooked, eg refs (2-7) and at least 2 meta-analyses (8, 9)”

Response: Thank you for the comment and all the suggested references. We have included some of the references and changed some others throughout the manuscript. We have not used all the references to reduce the total number of references in the text.

·     Point 24: “Lines 76-79: a point of difference in this study is that measures were taken in post-menopausal women.  This is a strength and should be emphasized in the purpose statement.”

Response: Thank you for the comment. We strongly agree with the suggestion and we have changed the statement in the manuscript.

 ·     Point 25: “Line 78: improve PEH is confusing.  To make it better means a further decrease in pressure?  Please ensure your hypothesis is clearly stated.”

Response: We have adjusted the phrase for better clarity and readability.

 ·     Point 26: “Line 95: “could not use” instead please explain that they were excluded or this was an exclusion criteria” 

Response: We have adjusted the phrase for better clarity and readability.

 ·     Point 27: “Line 116: “crossover, randomized, double blind, placebo-controlled” consider changing the order of these descriptors for clarity.”

Response: We have adjusted the phrase for better clarity and readability.

 ·     Point 28: “Line 117: 5 days washout with 3 conditions, 10 days doesn’t add up?”

Response: We have adjusted the phrase for better clarity and readability.

 ·     Point 29: “Line135: please consider a longer abstinence from nitrate rich foods in future studies”

Response: Thank you for your suggestion. We have used this duration based on some studies, however we did not find a standard recommendation about it. Unfortunately, there are several studies which have not controlled or mentioned this in their manuscript. However, we agree that a longer abstinence duration from nitrate rich foods would be better. 

 ·     Point 30: “Line 181: wording should be: statistical analyses were performed while researchers were blinded regarding the ingested beverage”

Response: Thank you for the comment. We have changed the sentence as suggested.

 ·     Point 31: “Table 1: some of your variables are n and % distribution, please add the heading where relevant”

Response: Thank you for the comment. We have changed the sentence as suggested.

 ·     Point 32: “Results: when a symbol separates two numbers, there should be a spaces on either side, eg 3.4 ± 3.0”

Response: Thank you for the comment. We have changed the sentence as suggested.

 ·     Point 33: “Line 197: wording “all sessions had similar values…” the sessions did not have values, rather the participants had similar values during the sessions”

Response: Thank you for the comment. We have changed the sentence as suggested.

·     Point 34: “Line 230-231: which resting measure was used?  There were four reported prior to the exercise.”

Response: We used the mean of the three measures that were performed at 9:00am, before the aerobic exercise performance. We have adjusted the phrase in the methods and results for better clarity and readability.

·     Point 35: “Line 245: “even though NO bioavailability was enhanced”.  This is debatable.  Salivary nitrite was elevated, but that does not necessarily translate into elevated NO bioavailability, which you have not measured.”

Response: Thank you for the comment. We agree with your point of view. We have adjusted the phrase for better clarity and readability. Based on previous studies, we believe that the ingestion of beetroot juice could increase both salivary and blood nitrite levels, and this could translate into elevated NO bioavailability.

 ·     Mcdonagh, S.T.J.; Wylie, L.J.; Webster, J.M.A.; Vanhatalo, A.; Jones, A.M. Influence of dietary nitrate food forms on nitrate metabolism and blood pressure in healthy normotensive adults. 2018.

·     Duncan, C.; Dougall, H.; Ohnston, P.; Green, S.; Brogan, R.; Leifer, C.; Smith, L.; Gowen, M.; Benjamin, N. Chemical generation of nitric oxide in the mouth from the enterosalivary circulation of dietary nitrate. 1995.

 ·     Point 36: “ Line 247: This is the main strength of your study.  Great!”

Response: Thank you for the comment. We appreciate it. We have changed some parts of the text to emphasize this information.

 ·     Point 37: “Line 248-255: as with the introduction, there are important citations missing in this paragraph.  This is not a thorough summary of where the current study fits within the literature.”

Response: Thank you for the comment. We have adjusted the phrase for better clarity and readability and have also include new references as suggested.

·     Point 38: “Line 256-257: please position references consistently, either after “review” and “meta-analysis” or at the end of the sentences, but avoid switching between these tactics”

Response: Thank you for the comment. We have adjusted the phrase for better clarity and readability and have also changed the references as suggested.

 ·     Point 39: “Line 258-259: opposing findings in the literature could be presented more clearly”

Response: Thank you for the suggestion. We have included more information in the discussion session.

 ·     Point 40: “Line 262-263:wording “collaborated to mitigate””

Response: We have adjusted the phrase for better clarity and readability.

 ·     Point 41: “Line 265: 12.9 (space) mmoles of NO (subscript) 3 – ”

Response: The sentence has been corrected as suggested.

 ·     Point 42: “Line 276: nitric oxide might contribute toward this effect, please be careful with overstatement”

Response: Thank you for the comment. We have adjusted the phrase for better clarity and readability.

 ·     Point 43: “Line 277: please avoid starting a sentence with an abbreviation”

Response:The sentence has been corrected as suggested.

 ·     Point 44: “Lines 284-285: Yes!  Well stated!”

Response: Thank you for the comment. We appreciate it.

 ·     Point 45: “Line 286: “wording: “justifying” ”

Response: We have adjusted the phrase for better clarity and readability.

 ·     Point 46: “Line 287: wording”

Response: We have adjusted the phrase for better clarity and readability.

 ·     Point 47: “Lines 287-293: Listing the possible mechanisms suggested by Halliwell suggests you are going to interpret these ideas and position your findings in context, you have not.”

Response: Thank you for the comment. We think it is important to point out these mechanisms even though we have not measured them. Therefore we have made adjustments to position our finds within the text.

 ·     Point 48: “Line 296: wording:“blockage or limitation”, please explain what you mean”

Response: We have adjusted the phrase for better clarity and readability.

·     Point 49: “Line 299-300: please read Forstermann, 2010 (10) and Selemidis, 2008 (11).  The relationship between oxidative stress and antioxidant capacity is misrepresented by this statement.”

Response: Thank you for the comment. We realized that there was a misrepresentation about this relationship. As we believe the most important information for the postmenopausal population is the fact that estrogen deficiency promotes endothelial damage, we have restructured the sentence to emphasize this point and solve the reported problem.

·     Point 50: “Line 302: “capacity” is incorrect here”

Response: We have adjusted the phrase for better clarity and readability.

 ·     Point 51: “Lines 305-309: this is an important statement for your paper.  Please ensure concise wording (avoid supposedly) and make your message very clear (what do you mean by difficult responses?)”

Response: Thank you for the comment. We have adjusted the phrase for better clarity and readability.

 ·     Point 52: “Lines 310-316: please read Burleigh et al, 2018 (12) and Blekkenhorst, 2018 (13), for details on the contribution of the oral microbiome”

Response: We appreciate your suggestion. We have restructured the paragraph to include more precise information regarding the importance of the oral microbiome by oral reduction of nitrate, as well as its clinical relevance for blood pressure control.

 ·     Point 53: “Lines 317-322: this is the crux of your discussion and should be discussed fully and referenced comprehensively”

Response: Thank you for the suggestion. Based on this and other suggestions made through the review, we have added more information and references in this paragraph.

 ·     Point 54: “Lines 327-329: these are author instructions that should have been deleted before submission”

Response: Thank you for the comment. We apologize for this mistake; the information has been deleted.

Round  2

Reviewer 2 Report

Dear Authors,

Thank you for your careful attention to my suggestions and recommendations.  I believe this version of your manuscript is substantially improved.

Congratulations on your good work!